# Physiological Action of Progesterone in the Nonhuman Primate Oviduct

**DOI:** 10.3390/cells11091534

**Published:** 2022-05-03

**Authors:** Ov D. Slayden, Fangzhou Luo, Cecily V. Bishop

**Affiliations:** 1Division of Reproductive & Developmental Sciences, Oregon National Primate Research Center, 505 NW 185th Ave., Beaverton, OR 97006, USA; luof@ohsu.edu (F.L.); cecily.bishop@oregonstate.edu (C.V.B.); 2Department of Obstetrics and Gynecology, Health & Science University, Portland, OR 97239, USA; 3Department of Animal and Rangeland Sciences, College of Agricultural Sciences, Oregon State University, Corvallis, OR 97331, USA

**Keywords:** progesterone, progesterone receptor, PRM, oviduct, nonhuman primates

## Abstract

Therapies that target progesterone action hold potential as contraceptives and in managing gynecological disorders. Recent literature reviews describe the role of steroid hormones in regulating the mammalian oviduct and document that estrogen is required to stimulate epithelial differentiation into a fully functional ciliated and secretory state. However, these reviews do not specifically address progesterone action in nonhuman primates (NHPs). Primates differ from most other mammals in that estrogen levels are >50 pg/mL during the entire menstrual cycle, except for a brief decline immediately preceding menstruation. Progesterone secreted in the luteal phase suppresses oviductal ciliation and secretion; at the end of the menstrual cycle, the drop in progesterone triggers renewed estrogen-driven tubal cell proliferation ciliation secretory activity. Thus, progesterone, not estrogen, drives fallopian tube cycles. Specific receptors mediate these actions of progesterone, and synthetic progesterone receptor modulators (PRMs) disrupt the normal cyclic regulation of the tube, significantly altering steroid receptor expression, cilia abundance, cilia beat frequency, and the tubal secretory milieu. Addressing the role of progesterone in the NHP oviduct is a critical step in advancing PRMs as pharmaceutical therapies.

## 1. Introduction

Several reviews address the action of ovarian steroid hormones on the mammalian reproductive tract [1,2,3,4,5,6]. It is well documented that estrogen (estradiol; E_2_) is essential for the normal development of reproductive tract organs and regulating cell proliferation and metabolism of tract tissues. Progesterone (P_4_) is also a crucial reproductive hormone with roles in the cyclic regulation of the fallopian tube [7,8,9,10,11], uterus [4,12], and cervix [13], as well as in supporting embryo implantation [14] and the maintenance of pregnancy [15]. Specific progesterone receptors mediate these actions, and synthetic progesterone receptor modulators (PRMs) are a class of drugs that alter P_4_ effects and have clinical utility as contraceptives and in treating P_4_-associated disorders [16].

The menstrual cycles of women and nonhuman primates (NHPs) are strikingly different from the estrous cycles of most laboratory animals [17]. Therefore, NHPs provide valuable animal models for translational studies relating to women’s health [18]. However, most literature reviews overlook the specific actions of P_4_ on the fallopian tube of NHPs. Therefore, our focus here is to describe the effect of P_4_ on the oviduct, highlighting the current state of knowledge on NHPs.

We begin this discussion with a brief overview of primate oviductal anatomy and histology during the menstrual cycle because the cyclic pattern of ovarian steroid secretion in women and NHPs is strikingly different from many mammals [19]. Our goal is not to discuss all historical studies, as these were addressed in previous reviews [4,11,20]. Like most mammals, primates have elevated levels of E_2_ in the follicular phase and elevated P_4_ in the luteal phase of the cycle. However, the primate corpus luteum also expresses high aromatase levels and secretes E_2_ in addition to P_4_ in the luteal phase [2]. Therefore, in primates, the cyclic changes in P_4_, not E_2_, drive cycles of epithelial differentiation in the fallopian tube (Figure 1).

## 2. NHP Oviductal Anatomy

The oviduct of women [20] and NHPs [4] is a tubular, seromuscular organ supported by the mesosalpinx, a portion of the broad ligament that anchors the uterus and oviduct to the body wall (Figure 2a). It is attached distally to the ovary and proximally to the lateral aspect of the uterine fundus. The fallopian tube consists of four anatomical regions based on tubular anatomy and distinct lumen histology [7]. These include the infundibulum (Figure 2a,b), ampulla (Figure 2a,c), and isthmus (Figure 2a,d), as well as the interstitial portion that passes through the myometrium wall. The infundibulum transitions into the ampulla, the longest part, and then to the isthmus of the oviduct. The inner mucosal layer of the tube is the endosalpinx, and the muscular layers are collectively called the myosalpinx. The lumen of the endosalpinx is continuous from the abdominal tubal ostium near the ovary to the uterine cavity.

The most distal portion of the oviduct, the infundibulum, is funnel-shaped with a fimbriated end surrounding the abdominal tubal ostium. Unlike rodents and domestic species, primates have no distinct ovarian bursa, and one of the fimbrial folds is usually attached to the ovary. The endosalpinx of the infundibulum and ampulla is extensively folded (Figure 2b,c) and contains ciliated and secretory endothelial cells (Figure 3). The myosalpinx of the ampulla consists of a thin inner circular layer of smooth muscle surrounded by an equally thin longitudinal muscle layer. The oviductal ampulla transitions into the most proximal portion of the oviduct, the isthmus. The isthmus has a thick circular smooth muscle layer that is continuous with the uterine myometrium. The cellular integrity of the endosalpinx in primates is dependent on ovarian hormones; ovariectomy results in almost complete atrophy of the endosalpinx epithelium of the infundibulum and ampulla. Interestingly, the isthmus portion of the tube is less responsive to hormonal cycles than the more distal ampulla and fimbria.

The ciliated cells of the endosalpinx and smooth muscle of the myosalpinx contribute to the movement of gametes along the oviductal canal [21]. Folding and hypertrophy of the fimbria and ampulla are dependent on E_2_. Co-administration of E_2_ with P_4_ in ovariectomized monkeys suppresses E_2_-driven tubal hypertrophy [11,22]. The ciliated cells of the infundibular fimbria create a current of peritoneal fluid toward the oviductal ostium, facilitating oocyte passage into the tube. The binding of spermatozoa to the ciliated epithelium occurs in many species, including primates [23,24,25,26], to provide a reservoir for fertilization [27,28]. In contrast to the ampulla, the isthmic and intramural epithelium contains only four primary epithelial folds and is strikingly less ciliated and therefore appears less sensitive to the actions of E_2_ and P_4._ Myosalpinx contractions move the oocyte to the site of fertilization in the ampulla and the developing embryo toward the uterus [29].

## 3. The Primate Menstrual Cycle

Old World nonhuman primates experience approximately 28–31-day ovarian cycles with prolonged follicular and luteal phases similar to women. These cycles are a striking contrast to many other mammalian species [17]. Laboratory rodents, for instance (e.g., mice and rats), display a short 4–5-day estrous cycle due to the formation of short-lived corpora lutea [30]; domestic livestock species (sheep, goats, and cattle) display estrous cycles featuring a short follicular phase followed by a long luteal phase [31]; and some mammals (e.g., rabbits) have induced ovulation [32]. Some species of New World monkeys do not reliably menstruate. However, Old World NHPs (e.g., apes [33], baboons [34], macaques [35], and vervet [36]) display actual menstruation with cyclic shedding of the endometrial lining [2,17,19]. By convention, the start of the cycle (day 1) is the first day of detectable menses, marking the beginning of the cycle’s follicular phase. The average luteal phase in macaques, baboons, and women is 10 days long, with P_4_ declining 24–48 h before menses.

Historical studies extensively characterized circulating concentrations of ovarian steroid hormones throughout the menstrual cycle in women and NHPs, including the vervet, macaque, baboon, and chimpanzee [17,37,38,39]. Assay technologies have undergone remarkable advancements since many of these early studies. Over the past 50 years, our research center has assayed E_2_ and P_4_ levels in macaques with techniques including radioimmunoassay [40,41], automated electro-chemoluminescent assays [42,43,44], and liquid chromatography–tandem mass spectrometry (LC-MS/MS) [45]. Comparing these methods reveals that LC-MS/MS can provide greater assay sensitivity and thus improve hormone detection at low concentrations. However, at normal cycling levels, the patterns described in early studies [17,39,43] remain reasonably accurate and are worth briefly describing here.

In macaques, early follicular phase (menstrual cycle day 1–8) serum levels of E_2_ average approximately 50 pg/mL and gradually rise to 100 pg/mL 2–4 days before the day of peak luteinizing hormone (LH) preceding ovulation. There is a surge in E_2_ to levels >350 pg/mL in response to the LH peak, followed by a rapid fall to about 25 pg/mL. Then, as the luteal phase proceeds, there is a second rise in E_2_ back to approximately 50 pg/mL. As the corpus luteum regresses, P_4_ and E_2_ levels fall to near the detection threshold for most assays (10–20 pg/mL). Human and chimpanzee E_2_ levels follow a similar pattern but trend slightly higher, with mid-luteal phase levels in the 100 pg/mL range [17]. This pattern of P_4_ secretion is very similar among all primate species.

Therefore, there is essentially a constant E_2_ > 50 pg/mL level in primates throughout almost all of the menstrual cycle, except for a brief mid-cycle E_2_ surge and a brief decline immediately preceding menstruation. Before ovulation in the cycle’s follicular phase, P_4_ levels range from 0.1 to 1.0 ng/mL. A small but significant rise in serum P_4_ coincides with the LH peak; then, as luteal formation occurs, levels in macaques rise to maximal values of 3–8 ng/mL. As observed with E_2_, in humans and chimpanzees, the luteal phase peak of P_4_ can be three to four times higher than in macaques. In all Old World primates, three days before the onset of menstruation, serum levels of P_4_ fall sharply to <1 ng/mL. This rapid fall in P_4_ triggers menstruation [46].

## 4. Cytologic Changes in the Oviduct during the Menstrual Cycle

Brenner and coworkers [11] rigorously examined the NHP oviduct in naturally cycling cynomolgus [35] and rhesus macaques [47] and compared the animals to ovariectomized monkeys treated with implants releasing E_2_ and P_4_ to produce controlled artificial cycles [48]. To characterize tubal morphology, they assessed epithelial cell height, the percentage of secretory and ciliated cells, and the abundance of mitotic cells and apoptotic cells, including the phagocytic macrophages containing apoptotic nuclear fragments. Cycle phase-associated apoptosis was also described for the epithelium of primates (macaques, baboons, and women) and non-primate mammals [49,50,51]. Brenner and colleagues examined the formation of cilia and cytologic features including extension secretory tips and the deciliation process in which the apical portions of the ciliated cells pinch off the cell bodies [52]. Their studies revealed that E_2_ and P_4_ are the only ovarian factors required to recapitulate regular cyclic changes in tubal histology identical to the natural menstrual cycle [4,11]. Ovariectomized animals displayed almost complete atrophy and loss of cilia in the epithelium of fimbriae and ampulla. Treatment of ovariectomized monkeys with E_2_ alone stimulated epithelial differentiation into a ciliated and secretory state. However, the sequential exposure to E_2_ followed by E_2_ + P_4_ resulted in epithelial regression to a non-ciliated and non-secretory state similar to ovariectomized, untreated animals. They defined eight morphological conditions associated with epithelial ciliation and secretory activity [11]. These stages are summarized in Table 1.

Figure 3 shows examples of the histology of the oviductal fimbria in the late follicular phase and the late luteal phase of the cycle. Luteal phase P_4_ acts as a master regulator of the cyclic oviductal differentiation against a background of continuous E_2_. At the end of the luteal phase of the natural menstrual cycle, most of the epithelium of the fimbria and ampulla is cuboidal with very few ciliated and secretory cells. Then, in the follicular phase’s post-menstrual period, epithelium hypertrophies become columnar and ciliated, and secretory cells develop to dominate in the infundibular fimbria and ampulla. These cells increase to a maximum height near mid-cycle and then shrink to a minimal height again by the mid-late luteal phase. Ciliated cells appear to shrink more rapidly than the secretory cells, and the apices of the latter are projected well beyond the tips of the cilia during the latter part of the cycle [22,53]. In ovariectomized animals, 2–3 days of E_2_ begin the process of oviductal differentiation into a ciliated and secretory state. The first evidence of deciliation and suppression of secretion often emerged within 48 to 72 h of the onset of P_4_ treatment.

The work of Verhage et al. [54] supported the view that the tubal epithelium of women undergoes cyclic changes similar to those of NHPs, and the epithelial cells attain their maximum height and degree of ciliation during the late follicular phase in both the fimbriae and the ampulla. It is noteworthy that some reports indicate a minimal change in the percentage ciliation during women’s cycle [55]. Those prior reports may not have fully appreciated the role of fallopian tube anatomy in the ciliogenic cycle and did not examine all tube sections. More recent studies in women confirmed these conclusions and added that an increase in epithelial mitotic activity occurred during the follicular phase when P_4_ was almost undetectable. Moreover, in macaques, the timing of the oviductal stages is not precise (Table 1). There is variability associated with tubal anatomy. These stages appear most intense in the oviductal fimbria, which is either the first to enter or the quickest to complete the ciliogenic process. The ampulla responds slightly slower to changes in the hormonal milieu when entering and completing the ciliated secretory state. When most cells near the lumen of the ampulla were regressing, a few ciliated and secretory cells could be identified that lagged behind the main population near the muscle wall.

Oviductal secretions have long been proposed to support fertilization and early embryo development [56]. These secretions and the regulation by steroid hormones in non-primate species have been recently reviewed [7]. Verhage and coworkers were among the first investigators to address steroid hormone-dependent oviductal secretions in primates [57]. Their studies reported the presence of secretory granules at the apical tips of secretory cells in the baboon. The same group then characterized an oviduct-specific glycoprotein (OVGP1) as an estrogen-dependent secretory protein synthesized by non-ciliated oviduct epithelial cells in various species, including macaque baboon and human [58,59,60]. While OVGP1 has recently been reported for other tissues, including the macaque cervix [61] and ovarian cancer [62,63], it remains a primary hormonally regulated secretory product of oviductal epithelial cells. OVGP1 appears to be an estrogen-upregulated protein that P_4_ and other pure progestins suppress, and pure PRAs reverse the effect of P_4_ treatment. During the normal menstrual cycle, OVGP1 is highly expressed in the secretory epithelial cells of the oviduct during the proliferative phase and is significantly reduced after ovulation [64]. In artificially cycled macaques, OVGP1 is reduced in P_4_ and levonorgestrel in the cervix [61] and oviduct. Expression of OVGP1 is a marker of P_4_ action and conditions including endometriosis that results in P_4_ resistance result in persistent oviductal OVGP1 expression [65]. Moreover, treatment of macaques with contraceptive levels of ZK137-316, a PRA compound similar to mifepristone, significantly increased OVGP1 protein in oviductal fluid [66].

## 5. Progesterone Receptors

Actions of steroid hormones to influence cellular function fall into two classifications: the slower classical genomic response and the rapid non-genomic response. The genomic actions of P_4_ in target tissues are mediated through interactions with intracellular progesterone receptors (PGR; also called PR), ligand-activated transcription factors that belong to the nuclear receptor family [67,68,69]. This family of transcription factors includes estrogen receptors (ER, ESR1, and ESR2), androgen receptors (AR), mineralocorticoid receptors (MR, nuclear receptor subfamily 3 group C member 2/NR3C2), and glucocorticoid receptors (GR, nuclear receptor subfamily 3 group C member 1/NR3C1). These are referred to as “classical” steroid receptors in which steroid binding leads to a long-lasting but slowly emerging response [68]. The transcription factor PGR is expressed in all P_4_-responsive organs, including the reproductive tract, mammary glands, cardiovascular system, and the oviduct [70,71,72,73,74]. To stimulate this “classical” response, binding of P_4_ to the ligand-binding domain of the PGR induces a conformational change that transforms the receptor from a static, non-DNA-binding configuration into one that activates gene transcription. This occurs by loss of associated (heat shock) proteins and dimerization of receptor moieties. The activated receptor–ligand complex can then activate the transcriptional machinery by direct action on regulatory motifs, most commonly at PGR response elements (PRE) sites, or by direct association of ligand-bound PGR with other transcription factors and coactivators [69,75,76,77,78].

Differences among fixation methodologies can greatly affect the outcome of ER and PGR localization by immunohistochemistry (IHC). For instance, studies of OCT-embedded cryosections of oviductal fimbria and ampulla of ovariectomized macaques revealed staining for ER and PGR localized to the nuclei of epithelial, underlying stromal cells and smooth muscles. However, staining of paraffin-embedded sections produced variable cytoplasmic plus nuclear localization. Since binding assays revealed that most of the ER and PR were recovered from the cytosol, IHC on cryosections was interpreted as indicating that PGR was rapidly translocated from the cytoplasm to the nucleus regardless of the hormonal state of the animal and interacted strongly with chromatin in the nucleus. It is noteworthy to mention that ESR-2 (ERβ) expression is reported for the oviduct of several mammalian species [7], but the role of ESR-2 during cyclic regulation in NHPs in unknown.

It is worth mentioning that PGR exists in two primary isoforms (A and B) encoded by a single gene but with different initiation sites that permit transcription of either a large or short isoform [71,79,80,81]. The larger (PR-B) isoform contains an N-terminal fragment of 164 amino acids that is absent from the short (PR-A) isoform. Thus, PR-B exhibits three transcription-activating domains (AF-1, AF-2, and AF-3), whereas PRA contains only two (AF-1 and AF-2) [82]. The two PR isoforms have similar steroid hormone and DNA binding activities but have distinct functions depending on the cell type and context of the target gene promoter. PRB appears to be a stronger transcription activator than PRA [80]. Due to the structural overlap of the two PGR isoforms, assessing the localization of the two PGR isoforms in the oviduct has been challenging. One approach is to use differential immunostaining with antibodies directed against PR-B and PR-A plus PR-B as well as specific differential PCR approaches. Using this approach, researchers at the University of Edinburgh reported attenuated PR-B in human fallopian tubes during the luteal phase of the menstrual cycle and during ectopic pregnancy [83]. However, cyclic regulation of oviductal PR-A/PR-B isoforms has not been confirmed in NHP studies.

Rapid, nongenomic actions of P_4_ are also reported for the oviduct. These are mainly attributed to so-called “membrane” receptors or “non-classical” progesterone receptors that appear to activate cellular second messenger pathways [68,84,85,86]. Among the rapid actions of P_4_ in the oviduct are the effects on ciliary beat frequency [87,88,89] and rapid alteration to spermatozoa motility [90]. Non-classical PRs include a family of membrane progestin receptors (mPRs) as well as the G-protein-coupled receptor (GPCR) family, which includes progesterone receptor membrane component (PGRMC), PGRMC1, and PGRMC2 [6,67]. The PGRMC family shares properties not associated with P_4,_ including a heme-binding domain related to some cytochromes. The mPRs were first reported in fish [85,91,92], and subsequently, five mPR subtypes (α, β, γ, δ, and ε) were identified [91] in a wide array of cell types in many mammalian species, including primates. The mPRs have no known homologies with GPCRs or nuclear PGRs, but are structurally related to adiponectin receptors and are classified as the progestin and adipoQ receptor (PAQR) superfamily. They display a predicted seven-transmembrane region and bind small steroid molecules, resulting in G-protein activation. However, the function of mPRs remains less clearly defined than that of the nuclear receptors [68]. This is largely due to a lack of data on the mPR steroid binding domains [93] and the absence of well-defined mPR modulators.

## 6. Cyclic Regulation of Steroid Responsiveness

It is well recognized that the oviduct is an estrogen-responsive organ that expresses ERα (ESR1) and PGR. Interestingly, ESR2 (ERβ) is also expressed in human fallopian tube ciliated cells, but the role of ERβ in NHPs remains to be determined. ER (not specific to ESR1 or ESR2) and PGR abundance have been assayed in naturally cycling NHPs as well as in NHPs treated sequentially with E_2_ and P_4_ to create artificial menstrual cycles. The earliest research characterizing the abundance of PGR in the primate oviduct utilized radiolabeled steroid binding on human fallopian tube [94]. These were followed by NHP studies that employed steroid binding and exchange assays to estimate levels of estrogen receptor and PGR (e.g., specific binding) in tissue homogenates [11,95]. These assays often used radiolabeled R2858 (a nuclear ER ligand) and R5020 (a nuclear PGR ligand) to avoid the metabolism of estrogen and P_4_. The sum of specifically bound steroids to the nuclear and cytosolic fractions from the homogenates represented an estimate of total receptor abundance. Binding of labeled R2858 and of R5020 were found to be significantly elevated in ovariectomized animals treated with E_2_ (or at mid-menstrual cycle) compared to hormone-depleted animals. This technique revealed that the oviduct’s differentiation into a fully ciliated and secretory endosalpinx epithelium was accompanied by significant increases in total ER and PGR [11]. Treatment of E_2_-primed monkeys with E_2_ in combination with P_4_ similar to the luteal phase resulted in significantly reduced levels of ER and PGR. In the case of ER, levels were reduced below those of ovariectomized untreated animals. Thus, ER and PGR expression were dependent on E_2_ action. Moreover, average ER and PGR levels were lower in animals treated with a combination of E_2_ and P_4_ than those observed in ovariectomized untreated monkeys. Because treatment with P_4_ alone failed to stimulate either ER or PGR, it was proposed that P_4_ acted to antagonize the effects of E_2_ on oviductal differentiation by suppressing ER levels below the threshold required to facilitate E_2_ action.

The overall relationship provided by classical binding assays appears to be more complex than was initially proposed. In concert with biochemical binding assays, cellular localization of ER and PGR by IHC on cryosections revealed that both cell and tissue type affected P_4_ suppression of ER and PGR. In support of binding assay results, the abundance of cells with strong nuclear staining for both ER and PGR increased in the follicular phase (and after E_2_ treatment) and decreased in the luteal phase (or after E_2_ plus P_4_ treatment). However, specific staining for epithelial ER and PGR were localized to the secretory epithelial cells, not the ciliated cells (Figure 3). This represents a paradox in that E_2_ and P_4_ strongly affect the ciliated phenotype, but staining is minimal in the ciliated cells. How can the dramatic effects of both E_2_ and P_4_ on the ciliated cells occur when the ciliated cells lack or express minimal receptors for both steroids? Moreover, PGR staining was almost completely absent in the epithelium during the luteal phase or after P_4_ treatment. This produces the question: How does P_4_ maintain its effects while suppressing its own receptor?

IHC revealed that strong ERα and PGR staining were present in stromal, smooth muscle, and secretory epithelial cells, suggesting that the effects of P_4_ on ciliated cells may be indirect. In the luteal phase (or after P_4_ treatment), ERα staining is retained in all the undifferentiated epithelial cells and in the underlying stromal cells, whereas PGR is minimal in the epithelium and retained (but noticeably less intense) in the stromal compartment. Therefore, one possibility is that the state of differentiation of the oviductal epithelium is mediated indirectly through soluble growth factors (or other unidentified mediators) secreted by ERα- and PGR-positive stromal cells. Moreover, stromal cells are separated from the epithelium by a definitive basement membrane, which could reduce the influence of soluble factors.

One potential mediator of P_4_ progesterone action, particularly in PR-negative cells, is the presence of specific mPRs or other non-classical PRs reported for human, murine, bovine, and canine oviducts, as well as ovarian cancers that may be of tubal origin. Oviductal mPR (beta and gamma) have been localized to bovine, human, and mouse ciliated epithelial cells [96] and may mediate the rapid effect of P_4_ on cilia beat frequency. However, localization of mPRs to oviductal cilia does not appear to reflect expected cyclic changes in ciliated cell abundance [96], as observed in nonhuman primates. Cyclic PGRMC1 and PGRMC2 expression and localization are reported for the macaque endometrium, but cyclic regulation in the NHP oviduct has not been extensively studied. The absence of reliable mPR/PGRMC modulators has significantly limited the study of these pathways in NHP models. In contrast, the expression and cellular action of nuclear PGR in the mammalian oviduct have been studied extensively.

## 7. Progesterone Receptor Modulators

The characterization of nuclear PGR isoforms has prompted the development of synthetic compounds called progesterone receptor modulators (PRMs) [97]. These include synthetic P_4_ analogs (progestins) and P_4_ antagonists (anti-progestins; PRAs) that bind to PGR and either stimulate or block PGR function [98,99,100]. It is noteworthy that long-term treatment with P_4_ and synthetic progestins reduces the abundance of ciliated cells in NHPs [101], and short-term treatment decreases cilia beat frequency in human oviductal cultures [102].

Mifepristone (RU486), the first well-characterized PRA, acts as a glucocorticoid receptor antagonist in the primate uterus, opposing various estrogen effects. The action of PRMs is often unique to the target organ, cell type, and sometimes the animal model examined. This has led to tissue-selective or physiologically selective PRMs [98]. The nuclear action of PRMs on classical (genomic) action of P_4_ has been most extensively evaluated because of the pharmaceutical potential of these compounds to treat gynecological disorders [97,98,99,103].

Compared to nuclear receptors, the action of PRM on mPRα, mPRβ, and mPRγ appears less clearly defined, with reports ranging from the minimal binding of mPRs to synthetic PRMs, especially classical PGR antagonists such as mifepristone [93,104], to putative or predictable actions [105]. Moreover, much of the studies on the nuclear action of PRMs have been conducted on non-primate models with strikingly different hormone profiles. In vivo assessments of mPR actions are confounded by co-expression of nuclear PGRs in many of the responsive cell types. However, differential binding of synthetic ligands offers the potential development of mPR-selective agonists and antagonists [93].

We have treated rhesus macaques with an array of potent PRAs including mifepristone [106,107], ZK 137 316 [108,109], ZK 230 211 [110], and CDB 2914 (Ulipristal) [111]. However, the primary experimental goal of these studies was to evaluate the action of these compounds on the uterine endometrium. Driving these studies was the observation that some PRA compounds, including mifepristone, have been reported to have unexpected anti-estrogenic actions on the endometrium. However, in the oviduct, pure PRA compounds such as ZK 230-211 appear to lack antiestrogen effects and block the genomic action of P_4_ [106,107]. In this condition, the estrogen action is unopposed, and the oviducts appear in a fully differentiated and ciliated secretory state.

PRMs can also provide contraception. Levonorgestrel is a contraceptive progestin that has inhibitory actions on oviductal cilia beat frequency and ovulation. Ulpristral and levonorgestrel have potential as emergency contraception, presumably by blocking ovulation. However, a secondary target may include oviductal cilia function. In vitro, progesterone can decrease human oviductal ciliary beat frequency (CBF) and muscular contractions, and the inhibitory effect of progesterone on CBF can be antagonized by mifepristone, a progesterone receptor (PR) modulator. Treatment of cycling rhesus monkeys with low-dose ZK 137-316, a compound very similar to mifepristone [112], prevented pregnancy at low doses that allowed menstrual cycles [113]. However, low-dose ZK 137-316 did not block ovulation and failed to alter oviductal differentiation and sperm passage but did significantly increase oviductal fluid levels of OVGP1 [66]. In contrast, ulipristal acts as a mixed agonist–antagonist compound and could disrupt gamete passage. This outcome on sperm passage may be PRA dose-dependent because other reports indicate that both ulipristal and mifepristone reduce ciliary beat frequency and contractility in human oviductal explants [114].

As indicated above, blockade of P_4_ action in NHPs is not associated with well-defined tubal abnormalities. Treatments with pure PRA, including ZK137-316 [112], mifepristone, and ZK 230-211, result in a fully ciliated and secretory tubal epithelium. This is not abnormal for the proliferative phase of the cycle. However, tubal abnormalities such as ectopic pregnancy are almost nonexistent in NHPs compared to women. Reproductive tract infections occur in NHPs and appear to be affected by estrogen and P_4_ action on the cervix, endometrium, and oviduct [115]. It can be speculated that treatment with mixed-action PRM therapy could alter normal cyclic changes. However, this represents a knowledge gap and further studies are required to assess the impact of P_4_ modulation on tubal dysfunction in NHP models.

## 8. Conclusions

The role of P_4_ in modulating oviductal morphology and physiology is indisputable. Cyclic changes in circulating P_4_ against a background of E_2_ stimulate changes in ciliary beating, muscular contraction, and oviductal fluid volume and composition, and, over time, suppress oviductal differentiation. These actions are mediated via intracellular nuclear receptors and via novel membrane receptors. However, the specific roles for membrane receptors remain to be resolved. Complicating these action mechanisms is that both classical nuclear receptors and fast-acting membrane receptors may be present in the same target cells. Thus, P_4_ can have rapid and long-lasting actions by stimulating paracrine factors that mediate hormone responsiveness. There is a large gap in our knowledge regarding P_4_-regulated effectors in the oviduct, although prostaglandins, endothelins, and growth factors may have roles as critical secondary mediators. Further development of selective PRMs that specifically target membrane receptors versus nuclear receptor isoforms may be required to elucidate the complex cyclic regulation of the primate oviduct.

## Figures and Tables

**Figure 1 cells-11-01534-f001:**
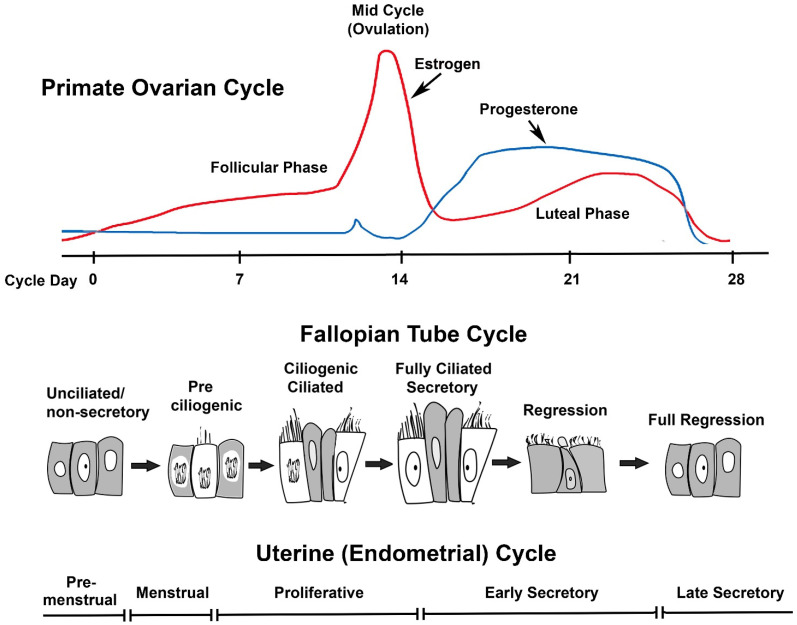
The fallopian tube cycle. Changes in the level of P_4_ in the presence of E_2_ drive cytologic changes in the fallopian tube epithelium. The upper panel shows an idealized primate ovarian cycle. In macaques, estrogen levels usually are 30–50 pg/mL at menstruation, rise during the follicular phase, and surge before ovulation. Post-ovulation, E_2_ declines but remains >50 pg/mL during the luteal phase. Progesterone (P_4_) levels are minimal in the follicular phase and rise significantly in the luteal phase. Estrogen levels ≥50 pg/mL drive cell proliferation in the pre-ciliogenic and ciliogenic/ciliated phases of the fallopian tube cycle and are necessary for maintaining differentiated ciliated secretory phenotype. The increase in P_4_ triggers regression of the epithelium despite continued E_2_ in the luteal phase. The lower panel depicts the phases of the uterine menstrual cycle, where the follicular and luteal phases are referred to as proliferative and secretory phases, respectively.

**Figure 2 cells-11-01534-f002:**
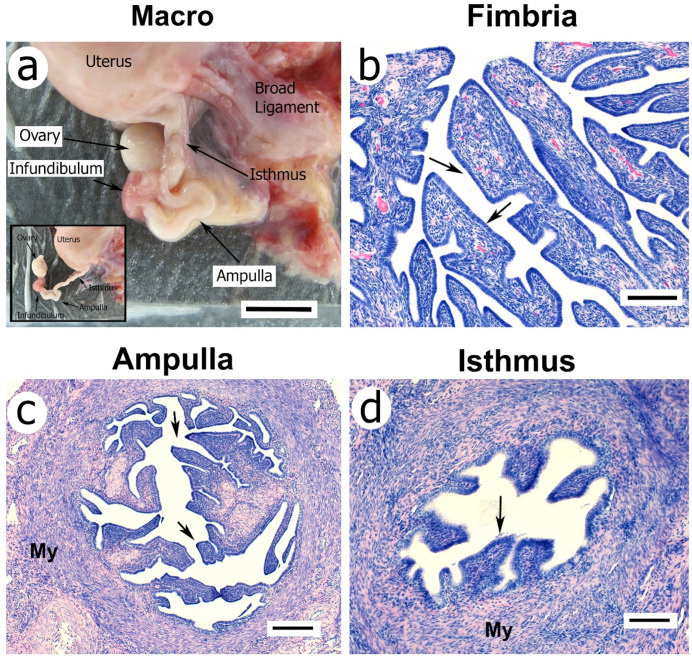
A photograph of the macaque reproductive tract (**a**) showing the uterus, oviductal infundibulum, ampulla, and isthmus. The inset shows the fallopian tube dissected free of peri-ovarian adipose and ligaments. Note the attachment of the fimbria to the ovary. Panels (**b**–**d**) show paraffin-embedded and hematoxylin-eosin stained (H&E) sections of the infundibulum fimbria (**b**), ampulla (**c**), and isthmus (**d**) from a rhesus macaque in the luteal phase of the menstrual cycle. Arrows show the cuboidal epithelium of the endosalpinx. My = myosalpinx. Scale bar in (**a**) is 1 cm. Scale bars in (**b**–**d**) represent approximately 100 μm.

**Figure 3 cells-11-01534-f003:**
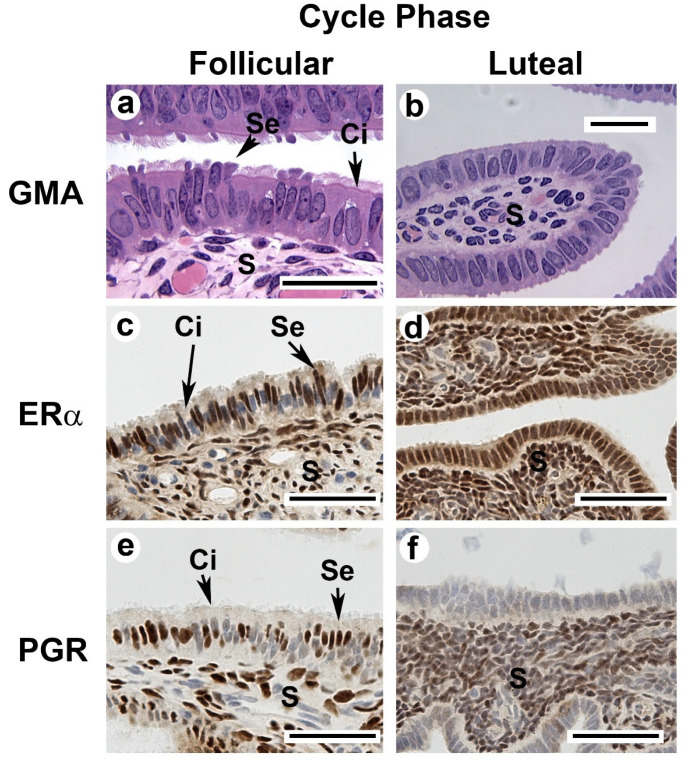
Photomicrographs of macaque oviductal fimbria collected in the follicular and luteal phases of the menstrual cycle. Glycol methacrylate sections stained with gills hematoxylin from the follicular phase (**a**) show differentiation into a fully secretory state. Luteal phase (**b**) is fully regressed (scale bars in (**a**,**b**) are 30 μm). In the luteal phase, presentation of cilia and secretion are blocked by P_4_. Immunostaining for ERα (**c**,**d**) and PGR (**e**,**f**) in cryosections revealed that the ciliated cells had minimal receptor staining in the differentiated state, whereas secretory cells stained strongly for both ERα and PGR. In contrast, sections from the luteal phase have strong ERα staining in all the cell types, whereas PGR staining was lost from the epithelium and retained in the stroma (scale bar in (**c**–**f**) is 50 μm). Together, this staining presentation implies that PGR regulation of oviductal differentiation may be indirect or mediated through non-classical mechanisms. Se = secretory cells; Ci = ciliated cells; S = stromal cells.

**Table 1 cells-11-01534-t001:** Cyclic stages of the oviductal fimbria in macaques.

Oviduct Stage	Uterine Cycle Stage	Description
Full Regression	Late Luteal/Pre-Menstrual	The epithelium is cuboidal with few ciliated or secretory cells. The epithelial cell nuclei appear shrunken.
Pre-Ciliogenic	Menstruation	This state is marked by epithelial cellular hypertrophy and mitotic activity. Epithelial cell nuclei swell, smoothing the nuclear contours.
Ciliogenic	Early Follicular	Cellular hypertrophy and mitotic activity continue. Histologically distinct light and dark cells can be identified, and cilia basal bodies are apparent in the apical cytoplasm of the light hypertrophied cells.
Ciliogenic Ciliated	Mid-Follicular	Mitotic activity has slowed; abundant ciliated cells and secretory cells are present. The word “ciliogenic” was placed first in the name of this phase to emphasize that ciliogenic cells predominate.
Ciliated Ciliogenic	Late Follicular	The majority of cells have become ciliated, but secretory cells have become much more prominent and have developed bulbous tips. The word “ciliated” was placed first in the name of this phase to emphasize that ciliated cells predominate over ciliogenic ones.
Ciliated Secretory	Periovulatory	Approximately 50% of the epithelial cells are ciliated. The remaining epithelial cells are secretory with prominent bulbous tips. There is minimal mitotic activity.
Pre-Regression	Early Luteal	This phase is similar to the ciliated secretory state. There is a striking increase in epithelial apoptotic cells, and macrophages are phagocytosing the apoptotic cells.
Regression	Mid-Luteal	The epithelium secretion is reduced, and the epithelium is undergoing deciliation. EM studies reveal ciliated cells to be pinching off their tips. Epithelial cell nuclei now appear shriveled. At the end of the luteal phase, the epithelium appears deciliated and cuboidal.

## Data Availability

Not applicable.

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
