# Peer review of "Physiological Action of Progesterone in the Nonhuman Primate Oviduct"

_cells, 2022, doi:10.3390/cells11091534_

Round 1

Reviewer 1 Report

The review discusses the physiological effects of progesterone in nonhuman primates' oviducts in very clear terms. The review cites and explains relevant findings in the literature clearly.   Although there are multiple spelling and grammatical errors. Proofreading can easily correct these minor mistakes.

Overall, the review is self-contained with relevant results to support the claims.

Suggestions:

  1. This review discusses how specific membrane receptors mediate the actions of progesterone, and how synthetic progesterone receptor modulators (PRMs) disrupt the normal regulation of the tubal cycle, altering the steroid receptor expression, cilia abundance, and the frequency of cilia beating, as well as the tubal secretory milieu. It would be helpful, however, if the review included an illustrated pathway map of the PRMs mechanism of action. It will help the reader to understand the review better and it will also make it more accessible to a wide audience.

Author Response

We thank you for the helpful suggestions.  We have proofed and updated the manuscript.  We have added a summary figure to the conclusions.

Again thank you for all your help.

Reviewer 2 Report

This is a well-written manuscript reviewing the role of progesterone on regulating cytological and physiological changes of nonhuman primate oviducts during the menstrual cycle and its mechanism of action. Only a few syntax errors were found as listed below.

Specific points:

  1. L125, P levels >> P4 levels?
  2. Figure 3, Scale bars?
  3. L299, please delete one “in”.
  4. L304, “…levels ER and PGR…” >> …levels of ER and PGR…?
  5. L321 & L328, excess space between words?
  6. L381, “… compounds including ??? have been…”.

Author Response

Thank you for your helpful suggestions.  We have proof read the manuscript and corrected each of your specific points.

1. L125, P levels >> P4 levels (corrected)

2.  Figure 3, Scale bars (Added)

3. L299, please delete one “in”. (fixed)

4. L304, “…levels ER and PGR…” >> …levels of ER and PGR…?  (Rewritten for clarity)

5.  L321 & L328, excess space between words? (Fixed)

6 L381, “… compounds including ??? have been…”.  (Rewritten for clarity)

Again, thank you for all your help.

Reviewer 3 Report

This review chooses primate as the model animal whose reproductive structure and menstrual cycle are more similar to humans compared to mammals, summarizing the secretion of estrogen and progesterone in the menstrual cycle, the structure of oviductal anatomy, the primate menstrual cycle, cytologic changes in the oviduct and the physiological effects of progesterone and their receptors changes on non-primate fallopian tubes. Authors point out that P4 is a key factor in regulating fallopian tube morphology and physiology. A few comments are listed below:

  1. There are many factors involved in regulating the fallopian tube cycle and ovarian cycle.Why did the author choose P4 as the main modulator in the review. Authors summarizes the fallopian tube cycle and ovarian cycle in primates, and shows P4, not E2, drive the fallopian tube cycle. However, previous study has shown that both P4 and E2 can regulate epithelial differentiation (Chen S, Einspanier R, Schoen J. Biol Reprod. 2013;89(3):54. doi: 10.1095/biolreprod.113.108829).
  2. The part of NHP oviductal anatomy can add a simple schematic diagrams, which describe the similarities and differences about the reproductive tracts of mammals, primates and human female.
  3. Authors should increase the contrast discussion of the effect of progesterone receptors and progesterone receptor modulators among mammals, primates and human female in Progesterone receptors and Progesterone receptor modulators sections.
  4. Authors should increase the review of the downstream signaling of Progesterone receptors sections. Progesterone receptor modulators sections should add the epigenetic factors.
  5. The review of pathological changes following dysfunction of progesterone signaling is insufficient.

Author Response

  1. There are many factors involved in regulating the fallopian tube cycle and ovarian cycle. ..... 

No.  There are not many factors that regulate the fallopian tube cycle.  The only ovarian factors required for cyclic changes in the oviduct are the steroid hormones estrogen and progesterone.  We point out that estrogen is required, but the cyclic changes in the primate driven by luteal progesterone.

2. The part of NHP oviductal anatomy can add a simple schematic diagrams, which describe the similarities and differences about the reproductive tracts of mammals, primates and human female.

No.  The purpose of this review  is to describe the effect of progesterone on the oviduct in nonhuman primates.  Not all mammalian species. We have stayed focused on the primate.

3. Authors should increase the contrast discussion of the effect of progesterone receptors and progesterone receptor modulators among mammals, .....i

No.  See above we are staying focused on NHPs 

4.  Authors should increase the review of the downstream signaling of Progesterone receptors sections. Progesterone receptor modulators sections should add the epigenetic factors.

No.  We are unaware of any epigenetic studies that have been done on progesterone regulation of the oviduct in primates.

5.  The review of pathological changes following dysfunction of progesterone signaling is insufficient.

We are unaware of documented oviductal pathologies associated with insufficient progesterone action in macaques or baboons. NHPs differ from humans in that cancers are rare.  It is possible that progesterone resistance leads to endometriosis but the role in the tube has not evaluated in macaques.

We are unaware of epigenetic studies that have been carried out on the oviduct of primates.

Round 2

Reviewer 3 Report

I have no further comments on the authors' responses from 1 to 4. As for the comment 5, although the authors mentioned that P4 is important regulator for oviduct cycle, but its modulation can have unintended effects on physiology of oviduct as well. They may mention these in their manuscript as a knowledge gap. 

Author Response

We have mentioned that this is a research gap.